# SUCCEED: Sharing Upcycling Cases with Context and Evaluation for Efficient Software Development †

**Takuya Nakata** [1,*], **Sinan Chen** [2], **Sachio Saiki** [3] **and Masahide Nakamura** [2,4]

[1] Graduate School of Engineering, Kobe University, 1-1 Rokkodai-cho, Nada-ku, Kobe 657-8501, Japan

[2] The Center of Mathematical and Data Science, Kobe University, 1-1 Rokkodai-cho, Nada-ku,
Kobe 657-8501, Japan; chensinan@gold.kobe-u.ac.jp (S.C.); masa-n@cmds.kobe-u.ac.jp (M.N.)

[3] Department of Data & Innovation, Kochi University of Technology, 185 Miyanokuchi, Tosayamada-cho,
Kami 782-8502, Japan; saiki.sachio@kochi-tech.ac.jp

[4] RIKEN Center for Advanced Intelligence Project, 1-4-1 Nihonbashi, Chuo-ku, Tokyo 103-0027, Japan

\* Correspondence: tnakata@es4.eedept.kobe-u.ac.jp; Tel.: +81-78-803-6295

† This paper is an extended version of our paper published in Takuya, N.; Sinan, C.; Sachio, S.; Masahide, N.
A Study of Case Sharing System for Efficient and Innovative Software Upcycling. In Proceedings of the
ICoDSE 2022, Denpasar, Indonesia, 2–3 November 2022.

**Abstract:** Software upcycling, a form of software reuse, is a concept that efficiently generates novel, innovative, and value-added development projects by utilizing knowledge extracted from past projects. However, how to integrate the materials derived from these projects for upcycling remains uncertain. This study defines a systematic model for upcycling cases and develops the Sharing Upcycling Cases with Context and Evaluation for Efficient Software Development (SUCCEED) system to support the implementation of new upcycling initiatives by effectively sharing cases within the organization. To ascertain the efficacy of upcycling within our proposed model and system, we formulated three research questions and conducted two distinct experiments. Through surveys, we identified motivations and characteristics of shared upcycling-relevant development cases. Development tasks were divided into groups, those that employed the SUCCEED system and those that did not, in order to discern the enhancements brought about by upcycling. As a result of this research, we accomplished a comprehensive structuring of both technical and experiential knowledge beneficial for development, a feat previously unrealizable through conventional software reuse, and successfully realized reuse in a proactive and closed environment through construction of the wisdom of crowds for upcycling cases. Consequently, it becomes possible to systematically perform software upcycling by leveraging knowledge from existing projects for streamlining of software development.

**Keywords:** software upcycling; software reuse; wisdom of crowds; collective intelligence; knowledge-based system

## 1. Introduction

*Software reuse* has been a topic of discussion since the first international conference on software engineering. For instance, during this conference, a study by McIlroy and colleagues proposed developing software routines in standardized reusable formats to allow for the recycling of software components across various projects [1]. Today, software reuse is realized and made available in various forms, such as software libraries, design patterns, and software frameworks, embodying typical software functionalities, practices, and architectures [2–4]. Moreover, research is advancing on the partial and small-scale reuse of existing software in other software developments through the reuse of code snippets [5,6]. Building on these achievements, as surveyed by Barros-Justo and others, modern software practices are increasingly leveraging reuse in various forms based on past assets [7]. Generally, development organizations possess developmental cultures

and practices that their members should adhere to, and reusing past projects can potentially streamline developments that align with these backgrounds [8,9]. Here, our reference to "organizations" encompasses a broad spectrum of groups, not limited to commercial application development sites or research institutions, which undertake various developments in the medium to long term and can cultivate a developmental culture and accumulate knowledge. However, there are cases where the background is not shared outside the organization and is not explicitly documented, existing only within the developed product [10]. The traditional open environment for software reuse is challenging in such situations, as no existing components are present, publicizing developmental assets is difficult, reusing specific small-scale elements such as libraries cannot extract context, and creating design patterns or frameworks in a closed environment is not cost-effective [11]. Traditional software reuse, whether in large-scale endeavors supported by many developers becoming frameworks or small-scale ones where developers search repositories for code snippets as needed, can be characterized as passive from the provider's standpoint. However, to reuse the development backgrounds that exist in a dispersed fashion within an organization, akin to a cloud, individual developers must actively suggest and share potentially reusable code snippets, architectures, and designs.

*Software upcycling*, which is a type of software reuse, is a concept involving the efficient production of innovative and high-value-added development projects using technical knowledge extracted from past projects as material [12]. Due to the inherent nature of upcycling, which involves extracting the core properties of development knowledge without decomposition, the knowledge available for reuse is not limited to code snippets. Depending on the upcycling method, materials that encompass rich developmental backgrounds, such as comments written in the code, architecture, design documents, development logs, and issues, can be targeted as well [13–16]. In upcycling, detailed analysis and materialization of specific development knowledge are conducted for reuse, enabling the utilization of open knowledge on the web as well as the incorporation of closed insights that are challenging to share outside an organization [17,18]. Furthermore, it encompasses a seeds-oriented approach aimed at maximizing the utility of existing technologies, making it suitable for active dissemination of valuable reuse materials by developers. Thus, software upcycling offers the potential for developers to actively communicate in a closed environment while targeting a wide range of materials and facilitating the reuse of organizational backgrounds, which is a challenge in traditional research but essential in the realm of software development. However, it remains unclear how to combine the material obtained from the project to perform upcycling.

In this study, *the Sharing Upcycling Cases with Context and Evaluation for Efficient Software Development (SUCCEED) system* is proposed as a method to streamline software development by utilizing the *wisdom of crowds* to aggregate upcycling cases performed by various developers. The "wisdom of crowds" refers to the phenomenon in which aggregating the opinions of a group of diverse individuals on a particular task can lead to better answers than those provided by individual experts [19,20]. For instance, in response to the question "what is the temperature in this room?", the average of guesses made by a group of individuals can be very close to the actual temperature measured by a thermometer. The key idea of the proposed method is to accumulate upcycling cases from various developers in a database as answers to the task of "how to upcycle by combining materials obtained from the project" and obtain the upcycling method that the searcher seeks through aggregation via searching and browsing cases. We conducted two experiments, one regarding upcycling cases and the other involving upcycling development using the SUCCEED system. By addressing three research questions (RQs), we aim to validate the effectiveness of software upcycling based on the wisdom of crowds.

A digest version of this paper has already been published at an international conference [21]. The most significant changes include the system implementation (Section 6) and the experiments and discussions (Section 7). We believe that these additions constitute a suitable contribution to a journal paper. The contributions of this study include the com-

prehensive structuring of technical and experiential knowledge beneficial for development, which was unachievable through conventional knowledge retrieval methods. Furthermore, we successfully actualized an innovative wisdom of crowds-based approach that allows for the reuse of intrinsic development contexts previously unattainable through traditional reuse methods. Additionally, we have verified the effects of collective intelligence on software upcycling. This enables software development to be streamlined by systematically utilizing existing project knowledge for software upcycling.

The remainder of this paper is organized as follows: Section 2 explains related research; Section 3 explains software upcycling and the wisdom of crowds as research backgrounds; Section 4 explains the goal of this study and the three research questions; Section 5 describes the architecture of the SUCCEED system; Section 6 explains the implementation of the system; Section 7 discusses the research questions and the proposed method based on two types of experiments; finally, in Section 8 we summarize the study and discuss future work.

## 2. Related Research

In the domain of software reuse, there is extensive research concerning the reuse of source code snippets. In research related to the reuse of source code snippets, Husasin et al. focused on a technique for searching related code using natural language queries [22]. They constructed a code search system by applying deep learning to open-source repository data. Their research focused solely on open-source code data, neglecting closed data and software materials other than source code snippets. In our study, we address the construction of a system that accumulates and searches for a broader range of software development knowledge. Abid et al. proposed a system for recommending code snippets tailored to specific development objectives, confirming that these recommendations can facilitate code reuse [6]. Their research targeted only the passive recommendation of source code snippets. In contrast, our research is devoted to the development of a system where various software materials can be actively recommended by developers. In a study by Papamichail et al., the authors proposed a method to objectively and statically assess the reusability of source code components in online resources, eliminating the subjective perspectives of experts [23]. They achieved objective evaluations by statically analyzing the source code. However, their method did not incorporate the subjective opinions of developers, which provide crucial perspectives on reusability. In our research, we integrate subjective opinions by aggregating collective intelligence, aiming to provide users with reusable software components.

In research concerning knowledge management in software development, there exist number of studies focusing on databasing and ontologizing of knowledge. Widyasari et al. constructed debugging tools by databasing bugs found in Python [24]. While their research was successful in leveraging insights from bugs rather than source code, the application of this knowledge remains confined to debugging. In our study, we do not merely consider bugs for reuse, instead aiming for their further application in various contexts beyond debugging. Martínez-García et al. proposed a process to ontologize knowledge related to architectures, and illustrated methods for knowledge reuse to prevent knowledge evaporation in software development [25]. While their research addressed the reuse of background knowledge, comprehensive software materials such as source code or domain models were not covered and their methods for leveraging knowledge remain vague. In our study, we are committed to building a model that targets a broader range of insights from software development and methods for its reuse.

Development based on past knowledge in software development is known as upcycling. The latest research based on historical knowledge includes methods for search and code generation using generative AI. As detailed in the work of Aljanabi et al., innovative search through natural language using generative AI tools such as ChatGPT is currently becoming feasible [26]. Yet, the reuse of past knowledge via ChatGPT is about passively reusing existing knowledge, not selective reuse of software projects, making it challenging to reuse insights that remain undocumented or potential development backgrounds that may not be well articulated. In our research, we focus on methods for selectively reusing

any software project. Additionally, as seen in studies by Biswas and Yetiştiren et al., code generation can aid in the software development process itself [27,28]. In addition to the issue of selective reuse with respect to past knowledge, as mentioned earlier, these studies face challenges in effectively incorporating domain-specific knowledge from development organizations. In our research, we endeavor to realize software reuse based on texts written by developers within an organization, aiming to leverage domain knowledge for reuse.

## 3. Preliminaries

### 3.1. Software Upcycling

The concept of *software upcycling* refers to the application of the *upcycling* concept, often discussed in the context of environmental issues, to software engineering. Figure 1 shows the flow of the software upcycling process. Upcycling is the concept of creating completely new products by adding arrangements while utilizing the characteristics of the original materials [29–31]. For example, considering the reuse of discarded mega-sized solar panels, upcycling uses the characteristics of the solar panels in a different context from their original purpose rather than reusing or recycling them. Upcycling is the idea of creating new value by repurposing materials for innovative purposes, such as creating a stylish table by attaching legs to a solar panel.

Software upcycling is the idea of converting parts of existing projects into new and valuable software assets. As shown in Figure 1, valuable functions and designs are retained and used in new projects while discarding implementation details. Materials used for upcycling are not limited to open resources available on the web; they can be extracted from software projects shared exclusively within development organizations as well. Existing exploratory studies report that careless reuse of online assets from platforms such as StackOverflow and GitHub can lead to issues such as bugs and increased development costs [5,32,33]. Therefore, leveraging closed assets, which are easier to validate for reliability, becomes crucial. The materials extracted from the original project need to be processed, such as by selecting important parts and modifying materials according to their purpose rather than directly integrating them into a new project. In addition, the implementation of reused materials and implementation specific to a new project are both necessary. Therefore, a process for creating upcycling recipes is required in order to consider how to process and use the extracted materials in achieving the entire project. However, recipe creation is a highly creative process, and is not easy.

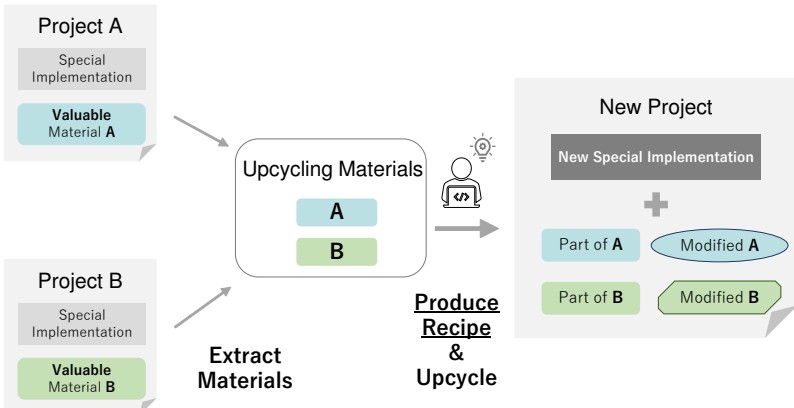

**Figure 1.** Software upcycling flow.

There are two approaches to software upcycling, namely, needs-oriented and seeds-oriented. The needs-oriented approach aims to upcycle existing knowledge created in existing projects to reduce redevelopment costs, while the seeds-oriented approach aims to combine various project materials to create innovative products. Thus, software upcycling is a useful technique for resolving technical debt as well as for finding value from projects that have become technical debt [34,35].

There are two challenges that should be addressed to improve the efficiency of software upcycling. One is the assetization and discovery support of project materials. Related research includes project corpus studies [12]. The other challenge is to discover how to use project materials and create software upcycling recipes. This study focuses on the latter challenge.

### 3.2. Wisdom of Crowds

The *wisdom of crowds* is a theory first proposed by Surowiecki; it suggests that the collective conclusion of a group when independently solicited for answers to a task among diverse individuals is better than the individual conclusions of the group [19]. For example, if a group is asked the question "what is the temperature in this room?" and each person predicts it without using a thermometer, there will likely be a range of predictions. In this case, averaging the individual predictions may produce a more accurate result than any individual prediction.

While an individual result may be considered random, it has been confirmed through follow-up experiments that the wisdom of crowds phenomenon can occur when appropriate conditions are met [36–38]. The four conditions that Surowiecki proposed are *diversity of opinion*, *independence*, *decentralization*, and *aggregation*. Diversity of opinion refers to the possession of unique personal information, including information that is not initially trustworthy. Independence is the ability to form an opinion without being influenced by others' ideas. Decentralization is the ability of individuals to utilize their specialized knowledge. Aggregation refers to the mechanism by which individual judgments are collected and combined into a single group judgment. The wisdom of crowds is not limited to numerical predictions of business performance, and can be applied to such varied fields as fact-checking of articles and facial recognition [39–41]. Surowiecki suggests that knowledge aggregation, such as that in Google searches, represents the wisdom of crowds, and research has applied the theory to knowledge aggregation related to software upcycling [19].

## 4. Goal and Research Questions

The goal of this research is to construct a method to systematically realize software upcycling and then verify whether the proposed method can enhance upcycled development. As a specific approach, we define development cases conducted through upcycling as upcycling case data models and propose *the Sharing Upcycling Cases with Context and Evaluation for Efficient Software Development (SUCCEED) system* as a repository for the wisdom of crowds in upcycling cases. In this study, we set three RQs, then conduct experiments using the proposed upcycling cases and the SUCCEED system to answer these RQs:

RQ1  What motivates developers to share upcycling cases?
RQ2  What characteristics are inherent in upcycled cases?
RQ3  How does collective intelligence enhance development in upcycling processes?

RQ1 concerns shared motivations, and addresses the challenge of externalizing knowledge accrued from past development projects which exists within the organization as software repositories, documents, notes, or undocumented individual and collective insights. In this study, we define upcycling cases as a format for making this development knowledge explicit. To address RQ2, we aim to investigate the characteristics of cases externalized in accordance with the upcycling case model with the purpose of clarifying whether a variety of insights useful for broad upcycling exists and whether the proposed model is suitable for representing this valuable knowledge. In this study, we design and implement the SUCCEED system as a platform for sharing upcycling cases. The objective of RQ3 is to determine how development is enhanced by using the collective intelligence formed by the externalized cases from various perspectives, such as development time and software quality. Considering that collective intelligence varies considerably depending on the people, cultural background, and knowledge accumulated within the forming organization, discussing the generality of collective intelligence in the context of RQ3 would be inappropriate. In this study, we conduct experiments on subjects in a case study environ-

ment at the Nakamura Research Lab at Kobe University, where the number of subjects is almost representative of the organization's population. Based on the experimental results, we address the RQs through contemplation from various quantitative perspectives, such as system usability and development time based on standardized quality characteristics, as well as based on in-depth consideration of software quality through analysis of recorded video and source code. Furthermore, we discuss whether the results can be generalized to a typical organization.

## 5. Methodology

### 5.1. Key Idea

We propose *the SUCCEED system* as a knowledge base for easily recording and sharing upcycling cases. The knowledge collected by the system consists of the personal opinions of the developers regarding the the question of "how to upcycle by combining materials obtained from the project", which is expressed as an upcycling case. The three items to be recorded and shared as upcycling cases are:

- Context, i.e., the purpose of the upcycling.
- Upcycling, i.e., the materials used and how they were upcycled.
- Evaluation, i.e., the positive and negative results of the upcycling process.

The SUCCEED system that accumulates the cases realizes the wisdom of crowds of upcycling knowledge and makes upcycling more efficient. Specific proposals include:

- Architectural design of the SUCCEED system.
- Definition of the upcycling material to be accumulated.
- Design of the upcycling case data model.
- Design of the SUCCEED system usage flow.

### 5.2. Architectural Design of the SUCCEED System

The architecture of the SUCCEED system is shown in Figure 2. The contribution side of upcycling cases registers the context, materials, recipe, and results of the upcycling through the web user interface (UI) operation screen. The system registers and manages the cases in the case database using the data model defined in Section 5.4. The receiving side can search for upcycling cases by keyword or contributor name through the web UI operation screen and obtain a list of cases. The contribution and receiving sides of the system are not limited to those who are familiar with upcycling, and are intended to include all developers. The system's architecture can function within a closed environment, allowing case sharing exclusively within an organization without any external publication.

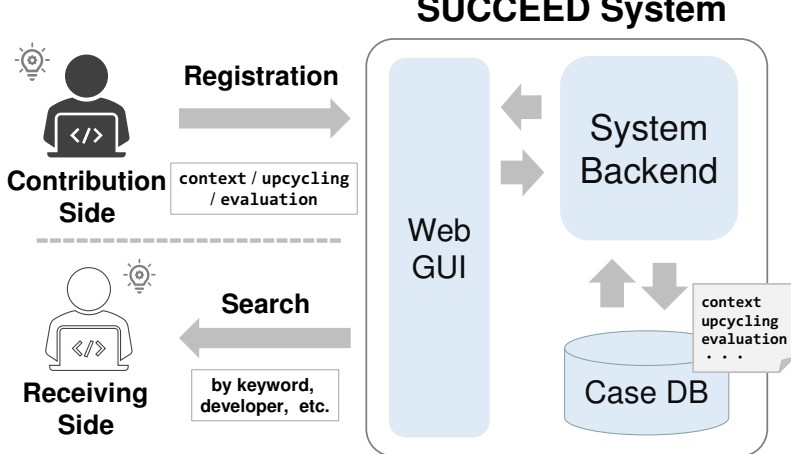

**Figure 2.** The SUCCEED system architecture.

### 5.3. Definition of Upcycling Material to be Accumulated

*Upcycling materials* are project materials used as materials for upcycling. As Figure 3 shows, a wide variety of project materials can be used, including the entire project, source code files, functions, application programming interfaces (APIs), test cases, and project design documents (e.g., Unified Modeling Language (UML) diagrams and data model designs) [42–44]. Limiting the types of upcyclable materials that can be accumulated in the knowledge base risks narrowing the possibilities for upcycling. Hence, the system aims to accumulate all kinds of project materials regardless of type or format. As a type-independent accumulation method, the system does not accumulates the upcycling materials themselves, only an overview of and access methods to these materials.

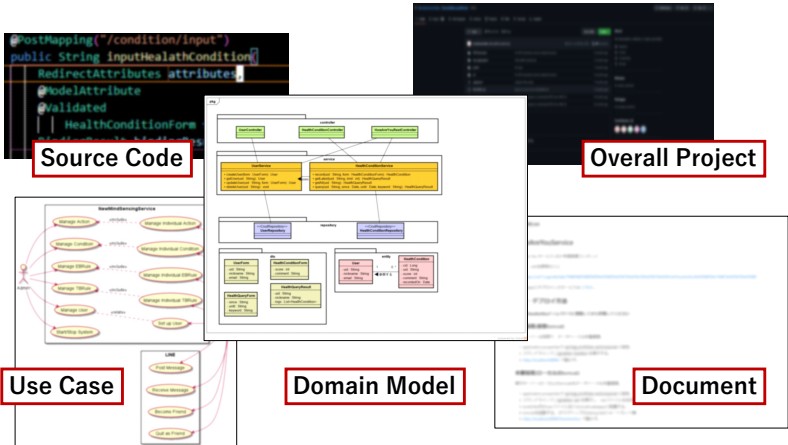

**Figure 3.** Different potential project materials.

One of the most important pieces of information in upcycling materials is the version [45,46]. Because project materials are at risk of being updated frequently, it is essential that the version of the material be noted and properly managed. Hence, upcycling materials that references a project without version control are of low value. Another important piece of information is how the upcycling material was discovered and acquired. There are two types of material discovery methods: in the empirical method, the developers discover upcycling materials from past projects they have been in charge of developing or know the upcycling materials from their actual upcycling experience; alternatively, a material search in accordance with the acquisition method can include the use of a project corpus [12] or a search for project materials.

### 5.4. Design of Upcycling Case Data Model

The upcycling case data model defined in this study is shown in Figure 4. The core elements of the upcycling case data model consist of the following three components:

- Context, i.e., the purpose and background of the upcycling activity.
- Upcycling, i.e., the materials used and the process employed for upcycling.
- Evaluation, i.e., the outcomes and assessment of the upcycling effort in terms of both positive and negative aspects.

*Context* is a crucial element that explains the purpose and circumstances behind the registered upcycling or the technical debts and challenges addressed through upcycling. Software upcycling always serves a specific purpose, with needs-based orientation aiming to reduce redevelopment costs and seeds-based orientation seeking to leverage buried technologies. Describing the context of each case enables users of the SUCCEED system to search for ideas in a way that is efficiently focused upon their specific objectives. The `context` component corresponds to the relevant aspect in the case data model. The `context` component comprises natural language summaries by registering the objectives, background, and challenges addressed by upcycling. The `context` component is deliberately

unstructured, allowing the registrants to articulate the development background through an intellectual process. This approach aims to manifest knowledge that has previously remained tacit while imposing minimal constraints and allowing for maximal expression.

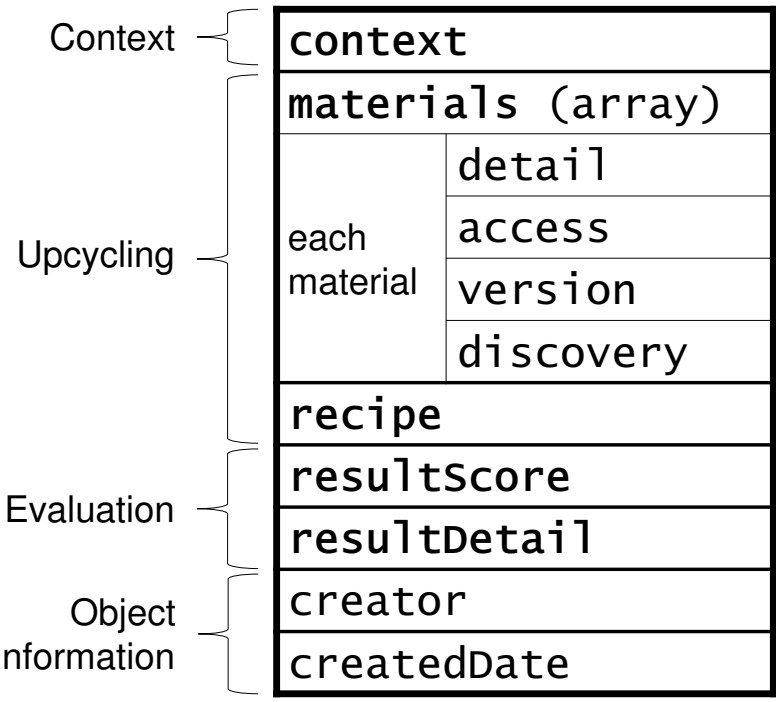

**Figure 4.** Upcycling case data model.

*Upcycling* elaborates on the specific techniques employed in the process, and comprises the `materials` and `recipe` components. The `materials` component lists the upcycling materials used, while the `recipe` component describes how these materials were processed and upcycled in detail. The `materials` are represented as an array that lists the upcycling materials. As mentioned in Section 5.3, each material's elements, namely, the detailed description, access method, version, and discovery approach, are essential. Therefore, the following four elements are defined as sub-elements of each `materials` entry:

- `detail` provides a narrative describing the content of the upcycling material, clarifying whether each material constitutes source code, design documents, or other resources, followed by a comprehensive explanation of its specifics.
- `access` pertains to the storage location or URL for accessing the upcycling material.
- `version` provides the version information of the upcycling material.
- `discovery` provides details about the circumstances under which the upcycling material was discovered.

*Evaluation* comprises a subjective assessment report made by the registrant regarding the upcycling. The `resultScore` component rates the upcycling on a five-point scale, while the `resultDetail` component elaborates extensively on the successes and areas for improvement in the upcycling process. In principle, the `resultScore` ranges from 1 to 5 points, though cases involving unimplemented upcycling ideas are assigned 0 points.

The `recipe` component details the specific process by which developers convert and utilize the `materials` component to develop a desired outcome. It is compiled by the developer in natural language and includes specific development methods and innovative ideas for processing materials. This knowledge can be reused from both purpose-driven and material-based perspectives. As in the `context` component, it is essential for the developer to articulate this in text, which is why it is defined as an unstructured natural language element.

Additionally, the upcycling case data model contains *object information* apart from its three core elements. The `creator` component represents the registrant of the case and the `createdDate` component denotes the registration date and time. This object information serves primarily as search information for the cases, and provides added information about when and by whom a case was registered, contributing to the case's credibility.

Examples of upcycling cases are shown in Figures 5 and 6. Figure 5 shows a single-material upcycling that reduces the cost of redevelopment of user password authentication. Figure 6 shows an upcycling with multiple materials, finding useful test cases from seemingly unrelated projects.

```
{
 "context": "Implement user password authentication for Project B using Java.",
 "materials": [{
   "detail": "Auth.java file in Web Application Project A",
   "access": "htttps://aaa/v3/security/Auth.java",
   "version": "3.2.4",
   "discovery": "Taught by XX, Project A developer."
  }],
 "recipe": "User password authentication for web pages similar to Project A can be implemented
           by simply rewriting the file dependencies",
 "resultScore": 4,
 "resultDetail": "User password authentication has been successfully implemented, however,
                  there is a possibility to change the password encryption method to a more
                  secure one in the future.",
 "creator": "Project B Development Manager XX",
 "createdDate": "2023-02-02T15:00:00"
}
```

**Figure 5.** Authentication function upcycle example.

```
{
 "context": "More test cases for user profile update
            functionality in User Management Project E",
 "materials": [{
   "detail": "UserUpdateTest.java file in user management project C",
   "access": "htttps://ccc/v1/test/UserUpdateTest.java",
   "version": "1.4.3",
   "discovery": "Taught by Project C developer XX."
  },{
   "detail": "Lines 210-240 of the ProductUpdateTest.java file in Product Management Project D",
   "access": "htttps://ddd/v2/test/ProductUpdateTest.java",
   "version": "2.1.2",
   "discovery": "Found using the project corpus."
  }],
 "recipe": "Add test cases that can be applied to users from Material 2, with Material 1
           as the main axis",
 "resultScore": 5,
 "resultDetail": "Material 2 test cases related to product updates were also extensive and
                  helpful in finding bugs, although the target data model was different.",
 "creator": "Project E Development Manager XX",
 "createdDate": "2023-02-02T16:00:00"
}
```

**Figure 6.** Multiple types of test cases upcycling example.

### 5.5. Design of the SUCCEED System Usage Flow

The execution flow of upcycling using the SUCCEED system consists of four stages: search, inspiration, implementation, and sharing, as shown in Figure 7. In the search flow, developers search for cases that can assist their development tasks from the cases registered in the system. The envisioned search methods involve using technical keywords related to the development and contextual keywords about the development purpose. As a result of the search, users receive case data detailing how materials were utilized in past upcycling and under what circumstances. Searching with technical keywords is expected to yield cases relevant to the `materials` and `recipe` components, while contextual keywords might return cases pertinent to the `context` and `resultDetail` components. In the inspiration flow, while browsing the search result, developers can consider how to use existing materials and other helpful materials to improve development efficiency along with design methods to optimize development. In the implementation flow, developers work on actual upcycling development based on the efficiency improvement methods obtained in the inspiration flow. They can then record the materials and utilization methods used in the development. In the sharing flow, they summarize the results of the practice flow in an article and register it in the SUCCEED system. Even upcycling cases based on personal insights that do not relate to utilizing the system can be registered. The decision around which cases to register lies with the developers, thereby ensuring active knowledge

sharing. The proposed system aims for efficient upcycling, and operates by cycling through these four steps.

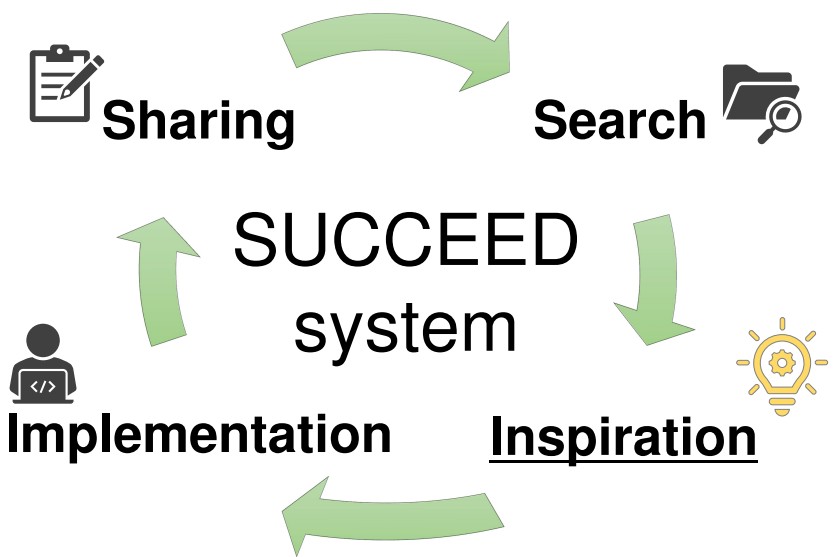

**Figure 7.** Upcycling flow.

### 6. Implementation

*6.1. Technologies*

The system implementation architecture is shown in Figure 8. Three servers were used for system construction. The first web server, Web Server 1, is a frontend server that implements the web user interface (Web UI) for user operations. It was implemented using JavaScript-based environments such as TypeScript, Next.js, and Material UI. The second web server, Web Server 2, serves as the backend server, facilitating internal system processes such as case registration. It was implemented using Java-based environments such as Kotlin, Spring Boot, and Tomcat. The Database Server is a server that realizes the databases of cases, materials, and comments in a relational database. The database management system employed for implementation is MySQL. We elaborate on the implementation details of the data models within the database. The case data model encompasses context, upcycling, evaluation, and object information. However, due to the nature of relational databases (RDBs), it cannot contain entities represented as arrays. Entities of each element in the `materials` component are stored in the material data model, and are linked to the case data model. The comment data model contains feedback comments from users to case registrants, and is associated with the case data model.

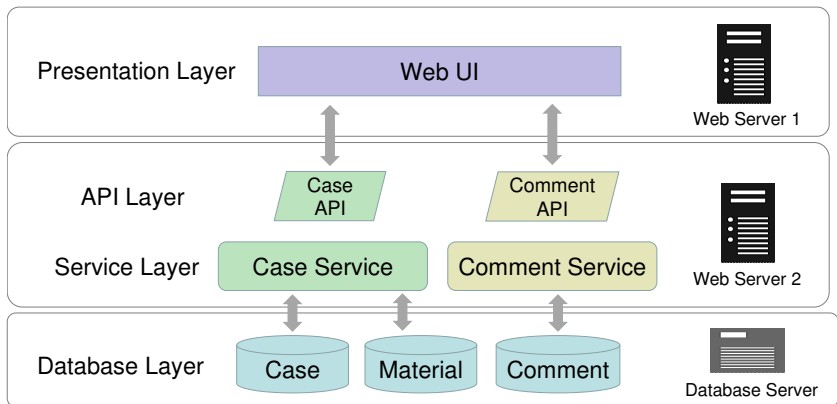

**Figure 8.** System implementation architecture.

The major characteristic of this architecture is the separation of the frontend and backend. Web Server 2 provides an Application Programming Interface (API) for case registration processing, while Web Server 1 operates by calling the API through the Web UI. The objective of providing the API is not to offer the backend functionality as a web service; rather, it is to segregate the frontend and backend. While the core of the SUCCEED system is the web service offered by the backend, the frontend is equally essential from a user experience (UX) perspective. Due to the different roles and responsibilities that the frontend and backend each hold within the SUCCEED system, it was deemed crucial to separate them clearly during implementation. Specifically, the servers for implementation and deployment were separated. Although it is feasible to further split the backend project into different microservices architectures, such as case services and comment services, the variety of backend functionalities is not very extensive; therefore, separating these roles and responsibilities does not provide very much advantage. For this reason, the backend internals were not segregated in this instance. The advantages gained from separating the frontend and backend are discussed in detail later. The server implementation, which includes the database server, can be hosted either on-premise or in the cloud. Even under constraints where there is not a suitable hosting environment within the organization or when internal knowledge cannot be placed in an external environment, it is possible to construct the SUCCEED system. Due to this separation, it becomes feasible, for instance, to operate the frontend solely using serverless cloud infrastructure. Moreover, the backend server's responsibility becomes clear regarding response time performance and scalability as the number of cases increases. On the other hand, the frontend server's responsibility lies in maintaining its function as the number of users and accesses grows. This can easily be managed by installing load balancers and scaling the frontend.

## 6.2. Web UI

The PC and smartphone versions of the Web UI screens are shown in Figures 9 and 10, respectively. The PC version of the UI design features a three-column layout for search, case display, and registration. The smartphone version prioritizes simplicity by initially displaying only cases for improved readability. Users can access the search and registration functions through a drawer menu, which can be invoked from the bottom right-hand button. In addition, the Good button function has been added for each case and comment function, enabling feedback from users to registrants. To assist with search and registration inputs, the help text explaining the concept of upcycling cases has been enriched.

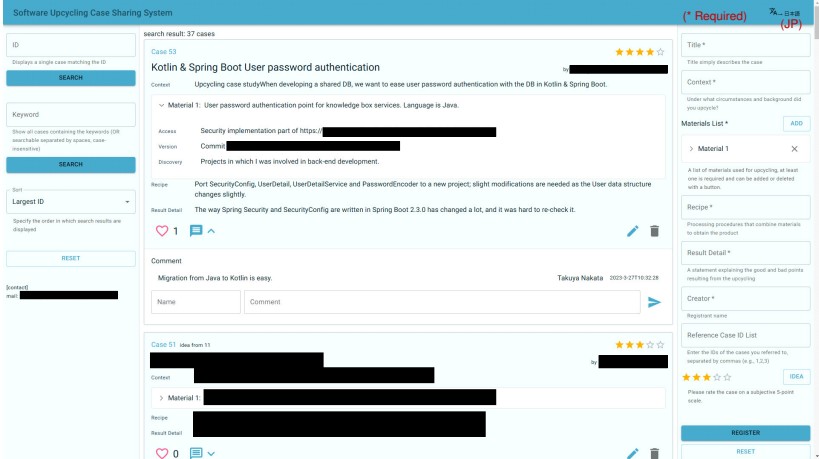

**Figure 9.** Web UI for PC.

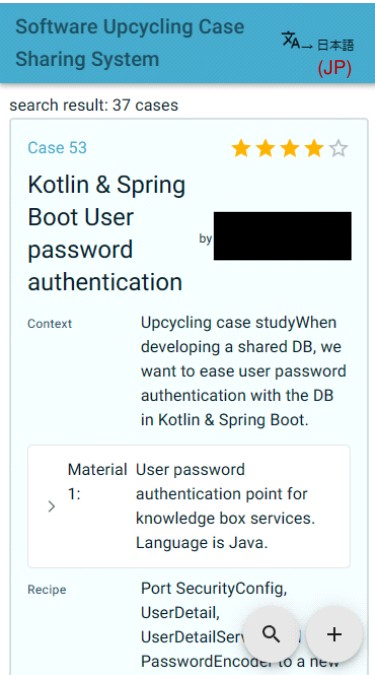

**Figure 10.** Web UI for smartphone.

## 7. Evaluations

### 7.1. Experiment 1

7.1.1. Experimental Setup

As the first experiment, we collected and analyzed cases in survey format without employing the SUCCEED system. The primary aim of this experiment was to obtain answers to RQ1 (what motivates developers to share upcycling cases?) and RQ2 (what characteristics are inherent in upcycled cases?). The cases collected in the experiment adhered to the format defined in Section 5.4 of the study. The survey participants consisted of seven students belonging to Nakamura Laboratory at Kobe University. The survey method involved having the participants fill out development cases they deemed relevant to upcycling in accordance with the case model structure. In the process of case submission, optional items could be left unanswered.

7.1.2. Experimental Results

Eight cases were collected, containing a total of ten materials. An overview of the cases is shown in Table 1. The types of materials were classified with duplication, and are summarized as a graph in Figure 11. Three materials had an empty version field due to the lack of version management. Furthermore, upon investigating the collected development projects against the laboratory's software repository, it was found that four projects did not exist in the repository, five projects that existed had almost no README documentation, and only two projects were present in the repository and had written README documentation.

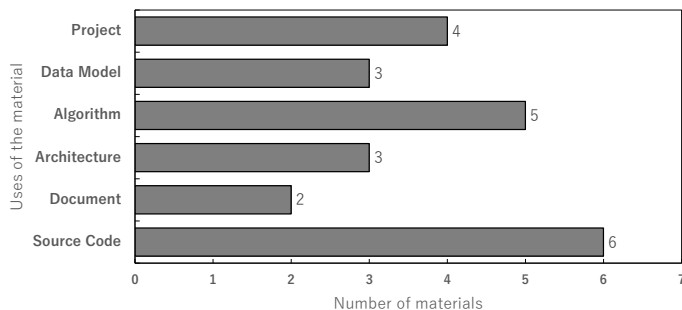

**Figure 11.** Classification of each material by type (allowing for duplication).

**Table 1.** Sample of cases from interviews.

| Summary | Result Score |
|---|---|
| The source code library dependency of the Pub/Sub service was used to develop a timer application using WebSockets. | 4 |
| The design architecture and source code of the knowledge sharing service using Spring Boot was used to develop the IoT infrastructure service. | 4 |
| To develop a scheduling service in GAS, algorithms from a Java service with almost identical functionality were reused. | 5 |
| To perform clustering analysis in Python, other clustering analysis code was reused and parameters were slightly modified. | 5 |
| To put a password on the diary service, the source code for the authentication was used, which was found on the internet. | 5 |

7.1.3. Discussion

Deliberating on RQ1, ten cases were obtained from the above experiment, as shown in Table 1. Each case represents a development instance chosen by the respondent from among their past projects that they deemed valuable to share with others due to its application of existing knowledge in an upcycling context. When asked about the rationale behind their case selection, responses included motivations such as the perceived need for upcycling and the difficulty in selecting previous projects that were not well-documented, leading them to opt for more recent projects that were fresh in memory. As displayed in Table 1, most of the cases gathered in the experiment had high outcome scores. When inquiring about the reason for sharing high-scoring cases, respondents mentioned not having detailed records of projects with poorer outcomes and a preference for showcasing positive results when sharing with others. While it remained unclear whether there were numerous high-scoring case studies available, it can be surmised that cases with more favorable outcomes were more likely to be shared. Based on these results, it can be inferred that when faced with the daunting task of knowledge documentation, developers are inclined to select cases that minimize psychological barriers and boost motivation by showcasing positive outcomes, along with those that are easier to recall. Furthermore, alignment with the overarching goal of collecting upcycling cases may serve as a crucial motivational factor.

Next, we turn our attention to RQ2. As illustrated in Figure 11, upcycling was not limited to source code, and included essential components of software programs such as algorithms and data models as well. Furthermore, upcycling was observed at the design level, that is, involving the overall project structure and documentation. This suggests that the process of collecting upcycled cases can contribute to the reuse of a broader range of project elements, surpassing the conventional reuse of code snippets commonly seen in previous studies. Our examination of the shared repository revealed that almost all of the shared cases derived from development projects that were not documented. This implies that the proposed model for collecting cases has the potential to capture a wide range of undocumented knowledge. Considering that several of the shared cases lacked version control, caution is necessary when considering the reliability of the included knowledge. Moreover, it was evident that the cases were from projects within the respondents' areas of

expertise and proficiency. Therefore, a distributed array of specialties among developers could lead to a richer diversity of registered cases. Summarizing these insights, the cases upcycled and gathered using the case model may encompass a broad spectrum of software materials, ranging from undocumented cases to those across diverse fields, all of which can be beneficial to numerous new development projects. Furthermore, factors such as version control and diversity among developers emerged as important considerations for enhancing the quality of the collected knowledge.

### 7.2. Experiment 2

7.2.1. Experimental Setup

As a second experiment, an analysis and discussion was conducted regarding actual development using the SUCCEED system. The primary objective of this experiment was to address RQ3 (how does the collective intelligence of the system enhance development in upcycling processes?).

The experimental outline involved assigning participants upcycling tasks and then determining whether there were any differences in development progression and output between the groups that uses the SUCCEED system and those that did not. The participants of this experiment consisted of students and faculty members from Nakamura Laboratory at Kobe University who were engaged in service development and data analysis. Excluding three co-authors, nearly all (a total of 11 out of 13) participated. Considering the small size of the overall population, it was believed that conducting discussions based on actual numbers and surveys, rather than relying on statistical tests, would be more reflective and pertinent to the entire laboratory population. This experiment did not use statistical tests, instead focusing on obtaining data closely resembling the entire population. Detailed time measurements, specific knowledge searches, development process records, and surveys were all undertaken. Subsequently, a comprehensive discussion and examination regarding RQ3 was conducted based on the results. For a comparison between those who used the SUCCEED system and those who did not, participants were randomly divided into two groups, Group 1 and Group 2, in nearly equal numbers. Each group switched between using and not using the system across two tasks, allowing for data collection from all participants during both usage and non-usage periods. This approach provides a foundation for discussions regarding the entire population based on the collected data.

The experimental procedure began by first standardizing the knowledge pertaining to the SUCCEED system. This was achieved through preliminary training on the system as well as practicing the search and registration functions. After that, the participants worked on two types of development tasks (Task 1 and Task 2) for 30 min each and the development process was recorded. After each task was completed, the development cases were registered in the SUCCEED system and the registration process was recorded. After all tasks were completed, the participants answered a questionnaire. In this survey, we established a set of questions to assess the effectiveness, efficiency, and trustworthiness using the Software Product Quality Requirements and Evaluation (SQuaRE) quality of use metrics [47,48]. Furthermore, the source code of the outcomes was submitted.

The difference between Group 1 and Group 2 was whether they used the SUCCEED system when working on their tasks. In Task 1, Group 1 was an experimental group that used the SUCCEED system at least once while freely using other search systems to develop, while Group 2 was a control group that did not use the SUCCEED system and only used other search systems to develop. In Task 2, the use of the system was reversed, with Group 1 as the control group not using the system and Group 2 as the experimental group using the system. To summarize the differences between the experimental and control groups, both groups followed a similar procedure for each task, consisting of four steps: search, inspiration, implementation, and sharing. The experimental group had the advantage of directly acquiring knowledge from the SUCCEED system during the search step. The control group could not directly obtain knowledge from the system; however, they could acquire similar knowledge from the web or other search systems that resembled the

information within the system. It was anticipated that the knowledge gained in the search step would influence the subsequent steps of inspiration, implementation, and sharing differently between the two groups.

The SUCCEED system used in the experiment was implemented as described in Section 6. The experiment was conducted with 24 upcycling cases already registered in the system before the start of the experiment. These cases represented sophisticated research activities in software development and data analysis input into the SUCCEED system by members of the same laboratory or those who had previously belonged to it. Of these, six cases were input by the experiment participants, with four people contributing one case each and one person contributing two cases; however, these particular cases were unrelated to the tasks. Although originating from the same organization as the participants, the intricate details of the implementations carried out by the other members were not well known to them. Due to the complexity of the case contents, most participants had little understanding of the registered cases prior to the experiment. The experimental tasks in this study were designed based on these preregistered cases. There was one case related to domain knowledge for Task 1 and one case each associated with development knowledge for Tasks 1 and 2. Moreover, several cases which were not directly related to any specific task included content pertinent to the technologies or the languages necessary for the tasks.

The computer used by the participants to work on the task was a personal computer that they were accustomed to using for software development on a regular basis. The computer's performance was adequate for development purposes, although not high-end. Ten participants used PCs with the Windows operating system, either Windows 10 or Windows 11, with a memory capacity of 8 GB or 16 GB, and the CPU was at most an Intel 10th generation. One individual carried out development on an iPad.

Task 1 involved the implementation of a mock input function for hand gestures in dialogue with a virtual agent [49]. The specified virtual agent was implemented in JavaScript. Knowledge about the specified virtual agent is closed and held exclusively by the Nakamura Laboratory at Kobe University; thus, obtaining detailed knowledge through open search tools such as Google was impossible. Methods for acquiring knowledge about the virtual agent included a closed research laboratory wiki, software repositories, and searching through the SUCCEED system. Two steps were required to implement the hand gesture function: Step 1 involved detecting various positions of the hand using a webcam [50], and Step 2 identifying the type of hand gesture from the hand position coordinates.

Task 2 involved the implementation of a simple algorithm for a shift automatic creation tool, with Python specified as the development language. The shift automatic creation problem is generally solved as an optimization problem. However, the closed knowledge held by the Nakamura Laboratory includes a simple algorithm similar to a greedy algorithm, which can only be accessed through the registered cases in the system. The simple algorithm has a difficulty level that can be easily thought of with basic knowledge about algorithms.

There were two features of the task settings. First, by separating the development languages of Task 1 and Task 2, the impact of changes in development speed due to short-term language specification familiarity was minimized. Second, the tasks set were not mere development tasks, as both involved upcycling as well. While the knowledge required for the development of Task 1 and Task 2 was closed knowledge registered in the system, this knowledge could be acquired without using the system. For Task 1, an understanding of the existing coding of virtual agents was necessary. The traditional way of writing code is not easily described as having a logical structure. However, changing the method would render it incompatible with existing systems. Instead of one developer changing the coding style, there is a need to upcycle the development without impacting the organizational culture. Thus, Task 1 is an upcycling task that involves adding functionality without redeveloping an entire system containing legacy code from scratch. For Task 2, ideas around modifying a shift automation algorithm to suit the task are required. While basic implementation is straightforward due to the availability of libraries for solutions easily found by web search, adjusting them can be challenging without mathematical knowledge. On the other

hand, solutions conceived of in a closed environment are simpler and easier to adjust; however, understanding them is challenging without strong code-reading skills. Task 2 involves reconstructing an algorithm suited to the task by referencing various existing algorithms; this is another upcycling task.

7.2.2. Experimental Results

We conducted measurements of various work times based on recordings of the development process and the registration of cases in the system for each task. The results are shown in the bar graph of Figure 12. In order to facilitate understanding, the experimental group and the control group are represented by white and gray bars, respectively, in both Task 1 and Task 2, as the groups were reversed between the two tasks. However, as seen with the data for Participant 1-C in Figure 12d, certain data were missing due to recording failures; these are indicated with an "x" mark. Additionally, only one participant each completed Task 1 and Task 2 within the stipulated 30 min time frame, making it impossible to obtain comparable task completion time data. However, we were successful in obtaining data on the completion time of Step 1, from the start of Task 1 to the successful detection of fingers. Figure 12a shows the results of measuring the time required to complete the finger detection subtask of Task 1. Figure 12b–d shows the results of measuring the time required to search for knowledge. The experimental group's time was measured using the SUCCEED system, while the control group's time was gauged using web searches. Usage of ChatGPT was not considered as a search, and was excluded. This issue is discussed in Section 7.2.3. However, for those who did not conduct each search, data could not be obtained; these cases are marked with a "y". Figure 12e,f shows the results of measuring the time required to register cases in the system for Tasks 1 and 2, respectively.

In the post-task questionnaire, responses were obtained for both Task 1 and Task 2 on the following items:

Q1   What knowledge do you already have that is required to solve the task (Task 1 options: hand recognition, overview of designated virtual agents, JavaScript, HTML, other free description; Task 2 options: shift creation algorithm, general understanding of algorithms, Python, other free description)?
Q2   Were you able to search for the target case using the system (options: yes, no, not used)?
Q3   Were you able to search for the target information using tools other than the system (options: yes, no, not used)?
Q4   Could you trust the content of the cases obtained as search results from the system (options: yes, no, not used)?
Q5   Could you trust the content obtained as search results from methods other than the system (options: yes, no, not used)?
Q6   What method(s) did you use to search other than the system (free description)?

From the results of Q1, almost all participants had prior knowledge of programming knowledge related to solving the task. Three participants were not familiar with JavaScript as used in Task 1, while all except one participant were able to use Python in Task 2 without any problems. The results of the questionnaire on the success or failure of the search in Q2 and Q3 are shown in Table 2. However, for users who did not engage in a search the determination of search success or failure remained elusive, necessitating their exclusion from the computation of the search success rate. Regarding Q4 and Q5, one participant answered that they could not trust the system's search method in Task 1. For Task 2, all participants expressed trust in the system's search results. All participants answered that they could trust the search results of the other search methods used in both tasks. From the results of Q6, all participants mentioned using Google search and ChatGPT as other search methods in addition to the SUCCEED system. Several participants used closed wiki or software repositories within the laboratory for searching as well. Additionally, one person indicated that they were able to perform the development solely using the SUCCEED system.

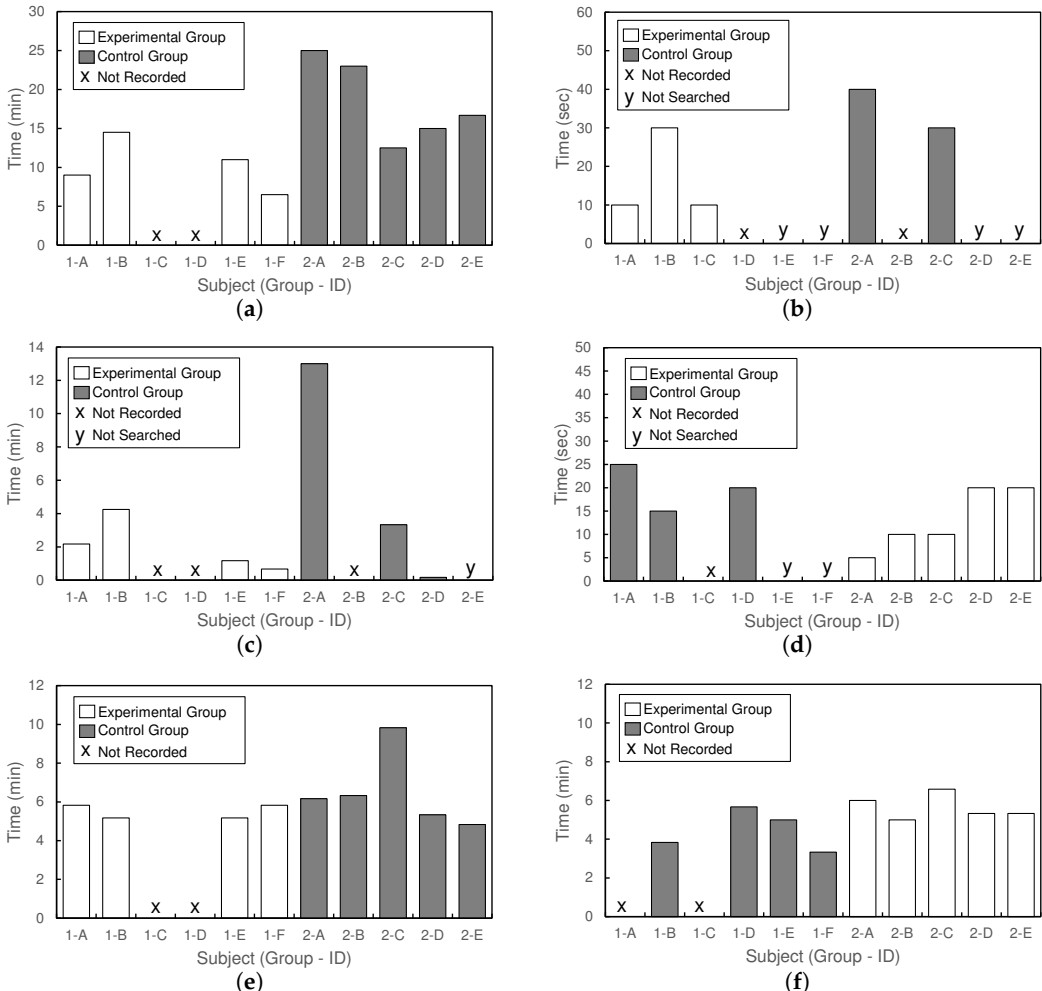

**Figure 12.** Results of time measurements: (**a**) time to detect fingers in Task 1; (**b**) time to search for agent in Task 1; (**c**) time to search for finger detection in Task 1; (**d**) time to search for shift algorithm in Task 2; (**e**) time to register the case of Task 1; (**f**) time to register the case of Task 2.

**Table 2.** Questionnaire results on success and failure in searching for task-related knowledge.

| Task | Search Tool | Success (Person) | Failure (Person) | No Search (Person) | Success Rate |
|------|-------------|------------------|------------------|--------------------|--------------|
| 1 | System | 6 | 0 | 0 | 100% |
| 1 | Others | 7 | 1 | 3 | 88% |
| 2 | System | 4 | 1 | 0 | 80% |
| 2 | Others | 8 | 2 | 1 | 80% |
| 1 & 2 | System | 10 | 1 | 0 | 91% |
| 1 & 2 | Others | 15 | 3 | 4 | 83% |

The following findings emerge from our examination of the video recordings and submitted source codes. For Task 1, Step 1, it was confirmed that all source codes functioned correctly. Successful finger detection was evident in the video recordings. In both tasks, source codes revealed snippets and structures referencing code from the web and the SUCCEED system. In Task 1, there was one participant who implemented without searching for the specifications of the virtual agent. As a result, they made an incorrect choice of the programming language. For participants who acquired knowledge of the virtual agent, the codes written for Task 1 adhered to the specified requirements and were applicable to the designated virtual agent. Both the experimental and control groups in Task 1 developed based on similar knowledge, resulting in functionally comparable source

codes. The primary differences lay in the process and time needed to acquire knowledge, and in a few cases the choice of implementation language. In Task 2, the experimental group aimed to implement a simple algorithm discovered through system searches, while the control group attempted to solve it using mathematical optimization that required mathematical knowledge which they found through Google searches on "shift automatic generation" and other similar search terms, resulting in completely different approaches to the task. The group that attempted to solve it using mathematical optimization failed to improve the source code within the experimental time frame due to a lack of prior knowledge. In Task 2, while the experimental group focused on essential coding related to the task, the control group worked on a generic shift generation problem and did not address the core issue of resolving the task. Concerning search functionalities, in terms of the freedom of search terms, such as allowing differences in capitalization and displaying similar results, other search services such as Google search are superior to the SUCCESS system, and there appear to have been several failures in searches using the system. On the other hand, there were many instances of people consuming a great deal of task time by browsing unnecessary pages due to long viewing times when confirming whether the discovered webpage was useful for the task, resulting in Google search failures. In addition to determining development guidelines, many people were observed using Google searches to obtain basic information about programming languages. In the development utilizing ChatGPT, there were instances where the exact content of the task was input directly. However, because ChatGPT does not possess organization-specific information such as details about virtual agents or greedy solutions, it produced irrelevant code. Two intriguing usages of ChatGPT were observed. First, there was one individual who let ChatGPT explain a case obtained from the SUCCEED system search, then input the task details into ChatGPT. This approach reflected knowledge from the case in the generated code. Furthermore, another participant developed in a different language and later used ChatGPT to convert it to the language specified for the task.

From the newly registered cases in this experiment, the following was revealed. In cases where knowledge from websites obtained through Google searches was used as materials, many only included the website URL and what was obtained through Google searches. However, a few cases included the search words used; the registration time for such cases was longer than the average time for subjects. For cases utilizing ChatGPT, the summaries of the materials simply mentioned "ChatGPT". The essential details about what kind of task was requested of ChatGPT were documented in the recipe section. When using websites or ChatGPT as source materials, there were no instances that noted any version-related details. Even in cases where tasks could not be completed, high scores were assigned to cases that were useful in the development process. In particular, cases where it was stated that the code was copied and used as-is tended to receive high scores.

### 7.2.3. Discussion

In deliberating on RQ3, the results obtained from the upcycling task in this experiment can be classified into three major steps: search, implementation, and sharing. First, based on the quality metrics during the use of SQuaRE, we assess the effectiveness, efficiency, and trustworthiness of the system's search functionality, which pertains to the search step. Next, we evaluate the effectiveness and efficiency of the system's registration feature, which is relevant to the sharing step. Subsequently, delving into the content of the implementation, which corresponds to the implementation step, we discuss the overall effectiveness and efficiency of the system from the perspective of its objective of assisting in upcycling. Lastly, synthesizing these viewpoints, we contemplate RQ3 more generally.

The effectiveness of the search function was evaluated based on the questionnaire results on success or failure of the search, shown in Table 2. The purpose of the search function in evaluating effectiveness is to enable users to obtain the knowledge that they want. Using the search function of the system, only one search failed out of eleven, resulting in a success rate of approximately 91%. In contrast, when using other search tools, three

searches failed out of eighteen, resulting in a success rate of approximately 83%. Therefore, it can be considered that the effectiveness of the system's search function is not significantly inferior to that of conventional tools such as web searches, wikis, or even the latest tools such as ChatGPT. However, this finding is predicated on the assumption that the system has accumulated knowledge that is beneficial for enhancing upcycling efficiency.

The efficiency of the search function was evaluated using the average time required by each group to reach knowledge through the search, with the results shown in Table 3. This outcome serves as a comparative analysis between the SUCCEED system and conventional web search. Although certain instances in the experiment utilized ChatGPT for developmental support, these were considered as independent variables and excluded from this assessment. From the results, it can be seen that the average search time decreased in all searches when using the system, indicating a certain degree of efficiency on the part of the system's search function. It can be considered here that much of the knowledge obtained through web search, which many subjects used, required time to confirm whether it was necessary knowledge, as there may not have been a succinct summary of how to utilize the knowledge.

**Table 3.** Average time required by group to obtain knowledge through search.

| Search Knowledge | Search Tools Used by Control Group | Average Time of Control Group (s) | Average Time of Experimental Group (s) | Average Time Difference (s) |
| --- | --- | --- | --- | --- |
| Agent | Software repository | 35 | 17 | 18 |
| Finger detection | Google search | 330 | 124 | 206 |
| Automatic shift generation | Google search | 20 | 13 | 7 |

The trustworthiness of the search function was evaluated by a questionnaire asking whether the search results were trustworthy. All respondents answered that they trusted the search results from outside than the system, while one respondent answered that they did not trust the system's search results. When interviewed, this respondent answered that they felt the knowledge found through the search was unreliable because there was no documentation on the material project referred to by the case found through the search. Therefore, they considered that the quality of case data, such as the level of detail in the description of each item of documentation and cases, affected its trustworthiness.

The effectiveness of the registration function was assessed by evaluating whether it achieved its objective of clearly organizing and summarizing cases in a structured format based on the content of the registered cases. Utilizing the SUCCEED system to register cases ensured that the upcycle cases for this experiment were organized and summarized in the model format. Even though there were variations in the amount of text for the `context`, `recipe`, and `resultDetail` components, which were described using natural language, no significant content was missing. However, regarding the versions part of the `materials` component, many omissions were observed in unmanaged sources such as websites, potentially leading to decreased reliability of the search function. Therefore, despite the deficiency in version information, it is believed that the registration function is effective in achieving its goal of clear organization and summarization.

By compiling the results from Figure 12e,f into the box-and-whisker plot in Figure 13, it was found that the average time for case registration was 5 min and 35 s, with a standard deviation of 1 min and 17 s. It was found that there was variation in registration time among subjects. In particular, cases with detailed descriptions for each item took longer to register, suggesting that efforts to improve the quality of the case may contribute to longer registration times, in addition to difficulties with case registration.

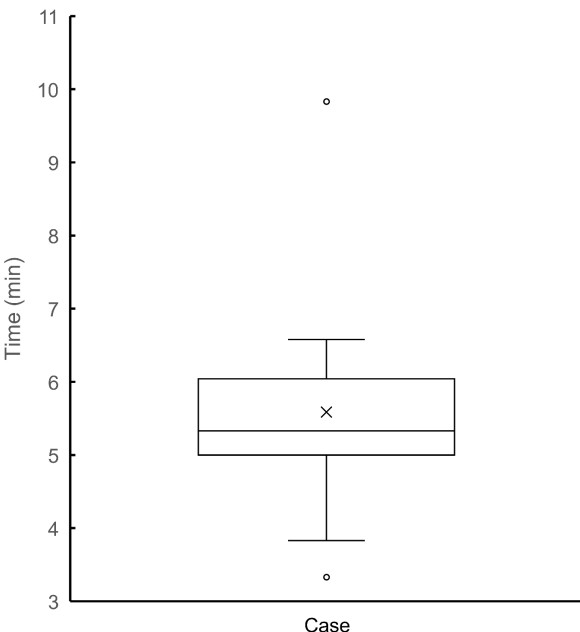

**Figure 13.** Box-and-whisker diagram of case registration times for all tasks.

To assess the overall effectiveness of the system, we discuss whether the SUCCEED system achieves its goal of aiding upcycling based on recorded videos during tasks and the source code of the outcomes. From the onset, observations from both the experimental and control groups' recordings and source codes confirm that upcycling took place during the tasks. This is evident from participants being seen acquiring knowledge through searches and ChatGPT and subsequently applying this knowledge with modifications in their coding. As stated in Section 3.1, upcycling involves leveraging the inherent characteristics of original materials while introducing alterations to produce an entirely new product. The developmental activities during the task align with this definition. When comparing the experimental group to the control group, both exhibited upcycling within the task and demonstrated functionality up to the correctly implemented sections. For Task 1, there were cases where the absence of domain knowledge concerning virtual agents led to developments that did not meet the task requirements. In Task 2, the approaches of the experimental group and the control group were entirely distinct; a more fundamental engagement with the core questions of the task was evident in the experimental group. The difference in knowledge acquired from the system's search function and that obtained from web searches or ChatGPT is believed to be the cause. As an effect of the system, not only can development time be reduced, it can potentially alter their development approach by conveying insights from upcycling as developers acquire necessary knowledge. From a software quality perspective, increasing opportunities to acquire essential knowledge through the system could make it easier for developers to accurately meet development requirements. Thus, the system has the potential to enhance the outcomes of upcycling through accumulated knowledge, rather than merely enabling upcycling.

We now discuss the system's impact on the efficiency of upcycling. From Figure 12a, the average completion time for the finger detection subtask in Task 1 was 10 min and 15 s for the experimental group and 18 min and 26 s for the control group. The average completion time for the experimental group was reduced by 8 min and 11 s, a decrease of 44%. These results were obtained from experiments primarily involving most members of the Nakamura Laboratory at Kobe University, the target organization for this study. In light of the difficulties in assuming a normal distribution for the average task completion time, which can vary significantly based on development skills and the developer's personality, a discussion based on actual measurements is more appropriate than a statistical discussion using tests. Therefore, in the experimental organization, the 44% reduction in development

time using collective intelligence suggests that upcycling utilizing collective intelligence effectively enhances development efficiency.

From the various analytical results presented here, we proceed to discuss RQ3. Initially, while users can quickly obtain the desired knowledge during the search step, it became evident that there was a need to register cases that were both comprehensive and reliable. In the implementation step, we observed a reduction in implementation time by 44% as well as the potential of improving product quality, such as meeting developmental requirements more easily and addressing core issues, due to the knowledge acquired via the system. In the sharing step, it was possible to consolidate cases in a format that other developers could refer to, which wasachieved within approximately 5 min and 30 s. Hence, it can be inferred that the SUCCEED system's upcycling enhanced development in terms of knowledge acquisition, efficiency, product quality, and knowledge sharing.

RQ3 was framed in order to investigate both the impact of the system and the influence of collective intelligence on upcycling. While the experimental group and the control group differed in terms of their use of the SUCCEED system for searching, this distinction does not necessarily equate to the use or non-use of collective intelligence. This is because, as Surowiecki has pointed out, commonly used search engines in modern development, such as Google Search, can be considered a form of collective intelligence [19]. Furthermore, because ChatGPT is a language model trained on various texts, it can be perceived as another embodiment of collective intelligence. Precisely speaking, the experimental group differed from the control group in its ability to utilize the SUCCEED system as a specific form of collective intelligence among others. From our experimental results, the characteristics of collective intelligence that enhance upcycled development become evident when contrasting implicit collective intelligence, such as web searches and ChatGPT, with explicit collective intelligence, as represented by the SUCCEED system. Implicit collective intelligence is shaped by knowledge not primarily shared for the purpose of forming collective intelligence, while explicit collective intelligence is formed by knowledge actively provided for that very purpose. This means that development is enhanced by collective intelligence characterized by its ability to actively disseminate structured knowledge and its selective assimilation of ideas from contributors, ensuring the gathered information is beneficial for upcycling.

This experiment was conducted in a specific environment, both organizationally and in terms of accumulated knowledge, at the Nakamura Laboratory of Kobe University. From the discussions to date, it is evident that in this environment the proposed system is beneficial for enhancing development. There is a high likelihood that its utility can be extended to a broader context. The insights accumulated within this experimental environment span a range of development styles, from individual to group projects, and cover a wide field including software development and data science. Thus, it can be assumed that the findings can be generalized and discussed in the context of various organizations. Based on the aforementioned discussions, we can conclude that the SUCCEED system, driven by collective intelligence, enhances development from the perspectives of knowledge acquisition, implementation, and sharing.

*7.3. Findings through Experiments*

Through this empirical investigation, it has been discerned that the establishment of an experimental methodology capable of configuring multiple subtasks and measuring task completion rates along with the attainment time of each subtask is advantageous. Due to the variability of task completion times arising from subjects' programming skills and knowledge, estimating task completion times proves challenging. Thus, the configuration of multiple subtasks proves beneficial. Furthermore, in the context of upcycling experiments, appropriate task configurations meeting the conditions of integrating previously unknown existing knowledge, registered as instances in the system with the necessary pre-existing knowledge for development, becomes imperative.

From the results in Table 3, it was found that the average search time varies greatly depending on the type of knowledge sought, such as hand detection and shift auto-generation, regardless of system usage. It is necessary to consider the differences in task achievement and completion time depending on the type of knowledge being searched.

Based on the combination of system and web searches in the experimental group, it is considered that the system is not a replacement for all functions of existing search methods, and is instead represents a development support tool to be used in conjunction with existing methods. In particular, due to the challenges in comprehensively covering the intricate specifications of programming languages in the SUCCEED system, the combined use with conventional online knowledge search systems via the web becomes essential.

### 7.4. Advantages and Limitations

The SUCCEED system proposed in this study achieves the wisdom of crowds by accumulating technical and experiential knowledge on upcycling in a group, and efficiently streamlines software development through upcycling. Furthermore, it has a great advantage in that it can efficiently upcycle based on limited knowledge that can only be shared within an organization, forming the wisdom of crowds. At the same time, upcycling is not limited to within an organization. By establishing an upcycling environment using the SUCCEED system across multiple organizations, there is potential for new upcycling utilization methods to naturally emerge. Additionally, because the system is designed for developers to proactively recommend their personal knowledge, there is potential to gather a vast amount of distributed and latent knowledge. Among the four conditions of the wisdom of crowds proposed by Surowiecki, diversity of opinions and decentralization are established by introducing practical cases through the practice flow, while aggregation is established by the aggregation mechanisms of the search flow and inspiration flow. As a result, it is believed that upcycling development cases using various project materials such as algorithms and architectures beyond just reusing conventional source code were collected, as shown in the experimental results in Section 7.1.

One limitation of this proposed method is that the independence of the accumulated knowledge in the system is not guaranteed, which may restrict the extent of efficiency improvement in upcycling. For the wisdom of crowds to function, respondents must answer independently; however, in the proposed method it is assumed that others will browse through their cases, leading to a decrease in independence. The key is how to encourage the registrant to register useful information that they themselves feel is helpful without being influenced by other cases when registering cases in the system, which can be addressed by enhancing the instructions. The knowledge aggregation mechanism, formed by the search flow and inspiration flow, significantly relies on the individual skill of the system user, which can influence the system's effectiveness. The search flow consists of two stages, namely, the search query and the result display. Both stages require refinement to enhance the overall accuracy of the aggregation mechanism. In this proposal, search queries are determined by matching keywords with words in the cases. To improve this, it would be possible to integrate more advanced research techniques such as fuzzy search in order to hit more candidates or recommendations based on development objectives described in natural language, as has been explored in code snippet search research [6,51]. For displaying the results, we propose a system that sorts by user-selected criteria such as registration date and highlights matching words. However, it is possible to consider introducing algorithms such as the commonly used PageRank to increase search accuracy. Moreover, the active sharing nature of the cases could set limitations on the system's knowledge collection capability [52], as developers need to be motivated to register and share examples. For instance, integrating gamification features such as contribution scores or rankings into the system might instill a sense of value and satisfaction in sharing, thereby encouraging more participation [53].

## 8. Conclusions

In this study, we have proposed a software development efficiency improvement method using software upcycling as a wisdom of crowds-based software upcycling case sharing system, along with its implementation and evaluation. The primary contributions of this research include: (1) The proposal of a case model that comprehensively structures technical and experiential knowledge beneficial for development. (2) The introduction of a system that systematically generates upcycling methods, marking the first endeavor to optimize upcycling. (3) Realization of software reuse through active knowledge sharing in a closed environment. (4) Elucidating how collective intelligence through the upcycling process can enhance system development. Through this study, we found that the wisdom of crowds in upcycling is achievable through aggregation mechanisms such as the search flow and inspiration flow of cases, and is useful for improving upcycling efficiency. However, the aggregation of development knowledge depends heavily on individual developer skills and motivation. Therefore, in the future it is imperative to employ classical techniques such as PageRank algorithms, gamification, and innovative search technologies based on natural language such as ChatGPT. These technologies can encourage developers to actively share knowledge and ensure smooth access to the accumulated knowledge. Future advanced research could focus on building systems that recommend effective upcycling cases derived from upstream development processes such as customer requirements and basic designs. These systems could estimate potential development costs as well. By accumulating and analyzing a vast amount of case data, we anticipate deriving new value-added information, such as understanding the causal relationships between context and evaluation in development and labeling materials based on context. While this study emphasized the human aspects of software development in its exploration of reuse, the added data obtained from case analysis may pave the way for further advancements in data-driven software engineering research.

**Author Contributions:** Writing—original draft preparation, T.N.; writing—review and editing, T.N.; supervision, M.N., S.S. and S.C.; validation, T.N. All authors have read and agreed to the published version of the manuscript.

**Funding:** This research received no external funding.

**Institutional Review Board Statement:** Not applicable.

**Informed Consent Statement:** Not applicable.

**Data Availability Statement:** The upcycling case data used in this study are unavailable because they contain information related to various products and studies.

**Acknowledgments:** This research was partially supported by JSPS KAKENHI Grant Numbers JP19H01138, JP20H05706, JP20H04014, JP20K11059, JP22H03699, JP19K02973, and Young Scientists (No. 23K17006).

**Conflicts of Interest:** The authors declare no conflict of interest.

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
