# Peer review of "SUCCEED: Sharing Upcycling Cases with Context and Evaluation for Efficient Software Development"

_information, doi:10.3390/info14090518_

Round 1

Reviewer 1 Report

The paper is good and easy to understand. The proposal is well presented and evaluated. However, to further improve the paper, I have some recommendations:

a) The research questions especially RQ1 and Rq2 looks more binary. It's better to reformulate them.

RQ1 Are there software upcycling cases within the organization that should be shared?

RQ2 If the upcycling cases exist, what kind of cases are they?

For instance, RQ2 is like a question for which you are seeking the answer provided a condition is satisfied. This could be taken as a research inquiry that seems not to be based on assumptions. It's better to make them non-binary.

b) Related work needs to be improved. Provide here systematic literature and then identify the research gaps.

c) The discussion section is too basic. Go beyond what you have identified and provide a discussion that opens interdisciplinary research doors. SImilary conclusion remarks need to be improved. 

d) Why you have repeated the REsearch questions (Introduction and Section 4). 

e) Some typos are there. Carefully check English and typos.

Some typos are there. Carefully check English and typos.

Reviewer 2 Report

The authors defined a model for upcycling cases and developed the Sharing Upcycling Cases with Context and Evaluation for Efficient Software Development (SUCCEED) system, using software upcycling as its wisdom. SUCCEED was implemented by utilizing upcycling cases performed by various developers. It is interesting and potentially useful. Moreover, the scientific component has several vulnerabilities and needs further work. However, a mechanism for estimating development costs and evaluating upcycling potential using existing knowledge needs further work.

I give some improvement suggestions:

1. From lines 147 to 150, "Surowiecki suggests...". Reference [16] or other references should be added.

2. Section 4 "Goal & Research Questions is too short (only 12 lines). It can be merged into section 5.

3. In section 5.3, according to Figure 4, the authors do not introduce "Meta Data" that include "creator" and "createdDate". However, "creator" and "createdDate" appear in Figures 5 and 6.

4. In line 262, "these four flows" should be "these four stages", according to line 250.

5. In section 6.2, in line 279, "The UI design features a three-column layout" should be "The PC version of the UI design features a three-column layout", according to Figures 9 and 10. However, the smartphone version of the UI has not been introduced.

6. In section 7.1.1, from lines 291 to 292, "The survey participants consist of seven students." What department and university did these students come from? (In section 7.2.1, from lines 320 to 321, "The participants in the experiment are six master’s students belonging to Nakamura Laboratory at Kobe University".)

1. In the paper, past tense (or present tense) is better than future tense. For example, from lines 335 to 341, the future tense means that the authors did not finish the survey and will finish it in the future. However, the past tense exists in line 341. There is no consistency.

Reviewer 3 Report

This paper addresses an interesting topic. I see this particular subject as becoming very relevant and papers like this one are important. The paper is well written and well structured. English use is good.
I understand that this is an extension of a previously published paper and sections 6 and 7 are very important for they are the new material. Most of my comments are on these two sections.

Section 5
Concerning the upcycling case data model
- the field "context" seems to be a very important component of this data. However, it is barely explained the context. Is this field is structured. It seems to be natural language? Shouldn't it have some sort of structure? How is it later queried?
- Same case as above: field "recipe" under field upcycling also seems important but is is also barely addressed in the discussion. Is this field a recipe for using the material in the upcycling? How is this recipe reusable when upcycling a given material in a different project/context? How is this structured and queried? Natural language?

Section 6
There are very few implementation details. It almost doesn't warrant a section at all. The UI screenshots are not particularly relevant in a journal paper and I was expecting other details. Examples of said details are:
- Discussion on the database layer implementation: data models (which and why).
- Performance and scalability discussion.
- Hosting options: own infrastructure or cloud based (is it possible?). Are there any constraints?
- API Layer implementation: is it web-services? what are the endpoints?
Because this is one of the two new contributions from the previously published paper, this section really should be more detailed.

Section 7
Concerning the experiment methodology, i have the following concerns
- The number of participants in experiment 2 is very reduced. It seems that it is only 3 participants for the test group and three more for the control group. I do not see strong reliability in any conclusion drawn from such small groups. Also, and perhaps because of this, there is no statistical processing of the results in the paper, which in itself is weird.
- It seems that RQ3 is addressed in the discussion in terms of time to find an answer from "google" vs SUCCEED. I find this troublesome, as RQ3 is not specifically about time (or only about time).
- Nowhere in section 7, or in the entire paper, is discussed the subject of quality of the software resulting from the upcycle. This is an important subject, more important than just metering the time needed to obtain answers from google/SUCCEED. I might assume that the software artifact was evaluated against a set of requisites, but it might also not be the case, and this is a big omission from the discussion.
- Another omission in the discussion is the subject of the previous knowledge base existing in SUCCEED. What knowledge is that, how was it inserted there, by who, and how does that knowledge relates to tasks 1 and 2?
- Concerning tasks 1 and 2, it is not clear how do they classify as a upcycle case. Given the description offered in the paper, it seems a simple task of programming, and not really upcycling preexisting artifacts. Perhaps it is the case that it was I that did not understood, but still, something seems to be missing.
- This study was conducted inside a given environment/institution. However, it would be interesting to discuss whether this type of approach can be used in an inter-environment (same knowledge base across different organizations/institutions).

As a final remark, I would suggest investing some thought and discussion about the way that information is processed within SUCCEED. The overall idea is that it is all text-based, natural language, and the processing is simple seeking for words. I do not know if SUCCEED goes beyond that. I would guess it would need to go beyond simple word matching in plain text, but that is a discussion for the authors to have, and me to read.

To summarize, I liked your paper and your line of research. There seem to be some missing things in the paper. It is my opinion that you should be given the incentive and opportunity to address the issues I noted before publication.

Round 2

Reviewer 1 Report

Most of my comments are addressed. However, still, major changes are required:

a) Why again you have one binary research question?

Refer to this: RQ3 Does the collective intelligence enhance development in upcycling processes?

b) Related work is still not very eloborated. Include references and better redraft it so that research gaps are clearly evident.

English is fine.

Reviewer 3 Report

It is noticeable that the authors did consider the issues I pointed out in the previous review. The paper was improved on several aspects and the authors letter of response was very clear on the improvements that were made.

I still have the following remarks.

1 - It was quite surprising to learn that new participants were added to the experiments. I do not imagine that authors had time and opportunity to conduct new experiments, i.e., new software development, compatible to the ones already existing, so I guess that there were some results from previous experiments that somehow were not considered in the first version. In any case, it is a bit odd, and it would be nice to have in the letter of response some explanation about these new results. Anyhow, there are a few more participants, but the total number remains low. However, I am not pressing this issue further.

2 - The statistical processing of the data remains to be done. However, the authors included new text explaining why it was not done making the results presentation a bit more convincing and coherent. Because of the low number of participants, I am not pressing this issue further, as it would require a new set of experiments – a new paper - not compatible with editorial timings. I am satisfied with the explanation given in the new paper text.

3 - Concerning the implementation details, it is worth noting the new text in this new paper version. However, the new details still leave much not described, and some of the new text is quite unclear. The foremost example is this “our research deemed it essential to provide an implementation that weakens their binding and distinctly separates their functional responsibilities.” – this is very unclear and adds not much to the reader. It seems that you opted for some sort modular (not monolithic) architecture, with low coupling between components. Well, this is the reader guessing and not actual facts given in the paper. And even so, what architecture is that? Micro-services? Other? Why and how? Concerning this I feel that improvements would still be needed.

4 – Regarding the issue concerning using mostly time to find answers as criterion for comparison: I read your new text and your reply, by it still bothers me that this remains the main criterion. According to this logic, one might arrive very quickly to a poor-quality solution and it would still count as an improvement. I also understand that in your experiments, apparently, all the solutions were deemed as good quality solutions and all requirements were fulfilled, but what we are talking about is a general methodology. I think there is room for improvement here.

5 – The new explanation about where the previous knowledge came from and its relation and applicability to the software developed during the experiments still leaves unanswered questions. The same can be said the way the quality of the resulting software was evaluated and its quality assessed. The new reasoning concerning why the experiments were indeed upcycling cases is also still lacking (yes, there is new text explaining, but it isn’t as clarifying as needed).

6 – The issue I pointed out earlier about applicability to new organizations and scenarios was probably not understood. I was referring to discussing about the feasibility of having a data repository that could be used by several organizations – basically, to what point is it possible to have common insights? And how would that be organized? This would be further work, and thus, I am not pressing this issue for this paper.

To summarize: when reading the new paper and your comments, I was left undecided between requesting a minor revision to address the new issues 3,4 and 5, or to accept it as is. They seem equally plausible. In such cases, it is my opinion that it should be decided in favor of the authors and that is what I am going to do. However, when preparing the camera ready, perhaps you can accommodate a few more improvements along the lines of what I pointed out in this new revision.
